# Current status of intestinal parasitosis and microsporidiosis in industrialized countries: Results from a prospective study in France and Luxembourg

**Maxime Moniot**[1,2], **Céline Nourrisson**[1,2,3], **Eloïse Bailly**[4,5], **Céline Lambert**[6], **Patricia Combes**[2], **Philippe Poirier**[1,2,3]*

1 Service de Parasitologie-Mycologie, CHU Clermont-Ferrand, Clermont-Ferrand, France, 2 Centre National de Référence Cryptosporidioses, Microsporidies et autres protozooses digestives (Laboratoire associé), CHU Clermont-Ferrand, Clermont-Ferrand, France, 3 Microbes, Intestin, Inflammation et Susceptibilité de l'Hôte (M2iSH), UMR Inserm/ Université Clermont Auvergne, Clermont-Ferrand, France, 4 Laboratoire de Parasitologie-Mycologie, CHU Dijon Bourgogne, Dijon, France, 5 Centre National de Référence Cryptosporidioses, Microsporidies et autres protozooses digestives (Laboratoire associé), CHU Dijon Bourgogne, Dijon, France, 6 Unité de Biostatistiques, DRCI, CHU Clermont-Ferrand, Clermont-Ferrand, France

* ppoirier@chu-clermontferrand.fr

**Data Availability Statement:** The authors confirm that all data underlying the findings are fully

## Abstract

### Background

Human intestinal parasitosis and microsporidiosis are a global health concern, mostly in endemic areas but should not be neglected elsewhere. Recent nationwide epidemiological data are scarce, especially from primary health care and developed countries. Diagnosis by molecular tools are increasing and several commercial gastrointestinal panel assays including protozoans and/or helminths are now available. These news tools improve the knowledge into real human parasite epidemiology. This study provides an epidemiological update on intestinal parasites found in primary health care in France and Luxembourg.

### Methodology/Principal findings

Two thousand fifty-six stools from primary health care patients were analyzed for the presence of intestinal parasites (IPs) during two different seasons of 2022, the winter and the summer, corresponding to more than 1500 patients from all over France and Luxembourg. Parasite detection was performed combining standard microscopy (merthiolate-iodine-formaldehyde and Bailenger concentration procedures) with two molecular panel assays (AMPLIQUICK Fecal Pretreatment, AMPLIQUICK Protozoans and AMPLIQUICK Helminths, BIOSYNEX, France). The prevalence of IPs in primary care patients reached 33.2%. *Blastocystis* sp. and *Dientamoeba fragilis* were the most frequently detected parasites in 20.5% and 13.1% of patients, respectively. Coinfection with two or more parasites was detected in 9.9% of patients. For some parasites, patterns according to gender, age, geography or season have been observed.

available without restriction. All relevant data are within the paper and its Supporting Information files.

**Funding:** The author(s) received no specific funding for this work.

**Competing interests:** The authors have declared that no competing interests exist.

## Conclusion/Significance

The high prevalence of pathogenic IPs (about 7%) underlines the importance of investigating gastrointestinal disorders through parasite examination, even in developed countries. The detection of parasites, pathogenic or not, remains a marker of the faecal-oral route of transmission and results should be interpreted accordingly. Parasites molecular characterization give new insights and should encourage further research as industrialized countries are not exempt of parasitic circulation and a better survey is necessary.

## Author summary

Recent nationwide epidemiological data on human intestinal microsporidiosis and parasites are scarce, especially among primary health care patients and in developed countries. Using novel data, we present an exhaustive epidemiology of intestinal parasites (IPs) in France and Luxembourg in 2022 from the primary health care patients. In this study, a high prevalence of IPs of 33.2% was found. *Blastocystis* sp. and *Dientamoeba fragilis* were the most frequently detected parasites, in 20.5% and 13.1% of patients, respectively. Considering only pathogenic parasites, the prevalence was 7.0%. Differences according to sex, region, season or age group were observed. Coinfection of two or more parasites was observed in 9.9% of patients. The use of molecular assays shows that IPs are underestimated in industrialized countries. The high prevalence of pathogenic IPs underlines the importance of investigating gastrointestinal disorders through parasite examination. The detection of parasites, pathogenic or not, remains a marker of the faecal-oral route of transmission and results should be interpreted accordingly.

## Introduction

Human intestinal parasitosis and microsporidiosis are a global health concern, mostly in endemic areas such as tropical countries, but should not be neglected elsewhere [1]. Recent nationwide epidemiological data are scarce in Europe, especially from primary health care patients. In France and Luxembourg, intestinal parasitosis surveys are not mandatory and so there is little epidemiological information, and the incidence of the infection is considered to be low. Most studies to date have centred on either extensive parasite research in a single center [2–4], or parasitic research in a wider geographic area (multicenter studies) but focusing on specific parasites [5,6]. Another limitation is the population studied, which is often a restricted category like children, inpatients or travellers, for example [2,3,7]. Epidemiological studies of primary health care patients are therefore needed to better estimate the prevalence of intestinal parasites (IPs) on a large scale and to understand parasite circulation in the general population.

The recent development of several commercial multiplex PCR panel assays and their prospective assessment have provided greater data on parasitic infections. These tools are more expensive than microscopy but can be used in routine laboratory practice and improve the sensitivity of targeted parasites [8–10]. However, the relevant studies suffer from the same limitations concerning population and certain parasite targets are still lacking for exhaustive results. In the present study, we performed an epidemiological update on IPs circulating in primary health care in France and Luxembourg in 2022. We combined standard microscopic

methods with molecular assays to obtain exhaustive epidemiological data and performed further molecular analyses for certain parasites such as species identification and genotyping.

## Materials and methods

### Ethics statement

The study was approved by local Ethics Committee (Comité éthique de la Direction de la Recherche Clinique et de l'innovation du CHU de Clermont-Ferrand) under number 2023-CF103 (IRB00013412, "CHU de Clermont-Ferrand IRB #1"). Due to the anonymity of the study, formal consent was not obtained.

### Clinical samples

Unpreserved stool specimens were prospectively collected during two periods in 2022, in the winter from February 13th to March 25th, and in summer from the 1st to the 30th of August. The samples came from primary health care laboratories that did not perform intestinal parasite analysis and outsourced it to our laboratory at the Clermont-Ferrand University Hospital Center. The stools were transported and stored at 4°C for two to five days prior to testing for the presence of ova, cysts, larvae, vegetative forms, adult stage helminths, and parasite and microsporidia DNA. The study population was therefore primary care patients with a prescription for IPs testing. Clinical data were not available, but the following patient information was provided with the samples: sex, age and ZIP code of residence. Patients were considered positive if at least one (or both) of the two diagnostic methods used was positive. Results are presented per patient: different stool sample results for each patient were pooled for analysis (i.e., reported as coinfection if different parasites were found in different samples).

### Macroscopic and microscopic examinations

All samples were prospectively analyzed by experienced operators in the Laboratory of Parasitology of the University Hospital of Clermont-Ferrand, France. For each sample, a macroscopic examination was performed to identify adult-stage helminths, followed by microscopic examination of fresh homogenized stools, and of merthiolate-iodine-formaldehyde (MIF) and Bailenger concentration techniques. In parallel to these analyses, each stool was prospectively aliquoted and kept at -20°C until use for retrospective molecular analysis.

### Molecular panel assays

At the end of patient inclusion, frozen samples were mechanically and chemically pretreated using the AMPLIQUICK Faecal Pretreatment assay (BIOSYNEX, France). Briefly, 200 mg of stool were ground with manufacturer's beads and then incubated at 95°C in lysis buffer for 5 minutes. DNA extraction of 200 μL of the supernatant was performed with the NucleoMag DNA Microbiome kit (Macherey Nagel, France) according to the manufacturer's instructions. The two molecular assays AMPLIQUICK Protozoans and AMPLIQUICK Helminths (BIOSYNEX, France) were used for the amplification of 20 different parasites, 10 protozoa and 10 helminths. Each panel included an internal control (to detect PCR inhibitors) and an in-process control (to validate the extraction and purification steps). All samples that did not satisfy each assay control from each panel were excluded from the analysis. The AMPLIQUICK Protozoans assay targets *Blastocystis* sp., *Cryptosporidium* sp., *Cyclospora cayetanensis*, *Cystoisospora belli*, *Dientamoeba fragilis*, *Encephalitozoon* spp., *Entamoeba dispar*, *Entamoeba histolytica*, *Enterocytozoon bieneusi* and *Giardia intestinalis*. The AMPLIQUICK Helminths assay targets *Ancylostoma duodenale*, *Ascaris lumbricoides*, *Diphyllobothrium latum*, *Enterobius*

*vermicularis*, *Hymenolepis nana*, *Necator americanus*, *Schistosoma mansoni*, *Strongyloides stercoralis*, *Taenia* sp. and *Trichuris trichiura*.

## Genomic analyses

To obtain full epidemiological data, species identification or genotyping was performed when the amount of DNA remaining was sufficient. All *Blastocystis* sp. positive samples were subtyped as previously described by targeting the small-subunit (SSU) ribosomal ribonucleic acid (rRNA) gene [11]. *Dientamoeba fragilis* genotypes 1 and 2 were distinguished by sequencing the SSU rRNA gene [12]. Species identification of *Cryptosporidium* was performed by an in-house PCR targeting a 258-bp DNA fragment located in the SSU rRNA gene [13]. Genotyping of *Cryptosporidium* (*i.e.* identification of subtype families and subtypes) was performed by next-generation sequencing of the highly polymorphic gp60 encoding gene as previously described [14]. *Giardia intestinalis* assemblages were determined by Sanger sequencing of the SSU rRNA gene as previously described [15]. Species identification of *Taenia* sp., *Hymenolepis* sp. and *Schistosoma* sp. were performed by sequencing the mitochondrial cytochrome c oxidase subunit 1 (COX1) gene [16]. *Encephalitozoon* species identification and *Enterocytozoon bieneusi* genotyping were performed by sequencing the internal transcribed spacer (ITS) region [17,18]. Finally, COX1 gene was sequenced to achieve species identification of *Sarcocystis* sp. [19].

## Definition of pathogenic or non-pathogenic parasite

The presence of parasites for which the pathogenic character is still debated was reported as carriage and were the following: *Blastocystis* sp., *D. fragilis*, *Entamoeba coli*, *Endolimax nana*, *Entamoeba dispar*, *Entamoeba hartmanni*, *Iodamoeba butschlii* and *Chilomastix mesnili*. All other parasites found in this study were considered pathogenic and their presence was reported as an infection.

## Statistical analysis

This cross-sectional study, conducted on a sample population, should make it possible to generalize the results to the entire target population. A margin of error on the estimate is defined to calculate the number of patients required in addition to the expected proportion of the primary outcome. Thus, for an expected proportion of intestinal parasitic carriage of about 50% (proportion requiring the highest number of patients), the inclusion of at least 1068 patients would make it possible to obtain an accuracy of ±3% of this proportion. Statistical analysis was performed with Stata software (version 15; StataCorp, College Station, Texas, USA). All tests were two-sided, with an alpha level set at 5%. No correction for multiple testing was applied in the secondary analyses [20]. Categorical data are presented as numbers and percentages, and age as mean ± standard deviation. The overall prevalence of intestinal parasitic carriage, and the prevalence of each parasite, were presented with a 95% confidence interval (95%CI), estimated by a binomial distribution. Factors associated with parasitic carriage (sex, geographical origin, season of sampling, and age) were studied with the Chi-squared test or the Fisher's exact test. For comparisons according to age groups, a post hoc multiple comparisons test (Marascuilo) was performed if omnibus p-value was less than 0.05. Multivariable analyses were performed for the two most prevalent parasites (*Blastocystis* sp. and *D. fragilis*) using logistic regressions and the following independent variables: sex, region, season, and age. The results are expressed as odds ratio and 95%CI.

## Results

### Study population

Overall, 2056 stool samples taken from 1589 patients were analyzed during the winter and summer periods (Fig 1). Twenty-two stools were excluded because of at least one invalid PCR control, leaving 2034 stools sampled from 1570 patients, made up of 1072 stools (840 patients) sampled in the winter and 962 (730 patients) in the summer. The 294 stools (283 patients) from Luxembourg were collected only during the winter period. Regarding seasonality, there was no difference between winter and summer in terms of age (42.7 ± 24.5 *vs* 44.9 ± 25.0 years old, respectively, p = 0.08) or sex ratio (male: 42.6% *vs* 42.9%, p = 0.92). The geographical origin of the samples (five ZIP code lacking) showed that more were collected from the North than the South, 1246 stools (983 patients) vs 781 (582 patients), respectively (Fig 2A). Ages ranged from one month to 99 years in a population comprising 276 children (*i.e.* <18 years old) and 1293 adults (one birth date was lacking) with a sex ratio male/female of 0.75 (n = 899 females, 57.3%, n = 671 males, 42.7%) (Fig 2B).

### Parasites detected

All parasites detected in this study by microscopy and/or PCR are listed in Fig 3. The overall prevalence of intestinal microsporidiosis and parasitic carriage was 33.2% (n = 521/1570 patients, 95%CI: 30.9 to 35.6). *Blastocystis* sp. and *Dientamoeba fragilis* were the most frequently detected parasites being identified in 20.5% (n = 322) and 13.1% (n = 206) of patients,

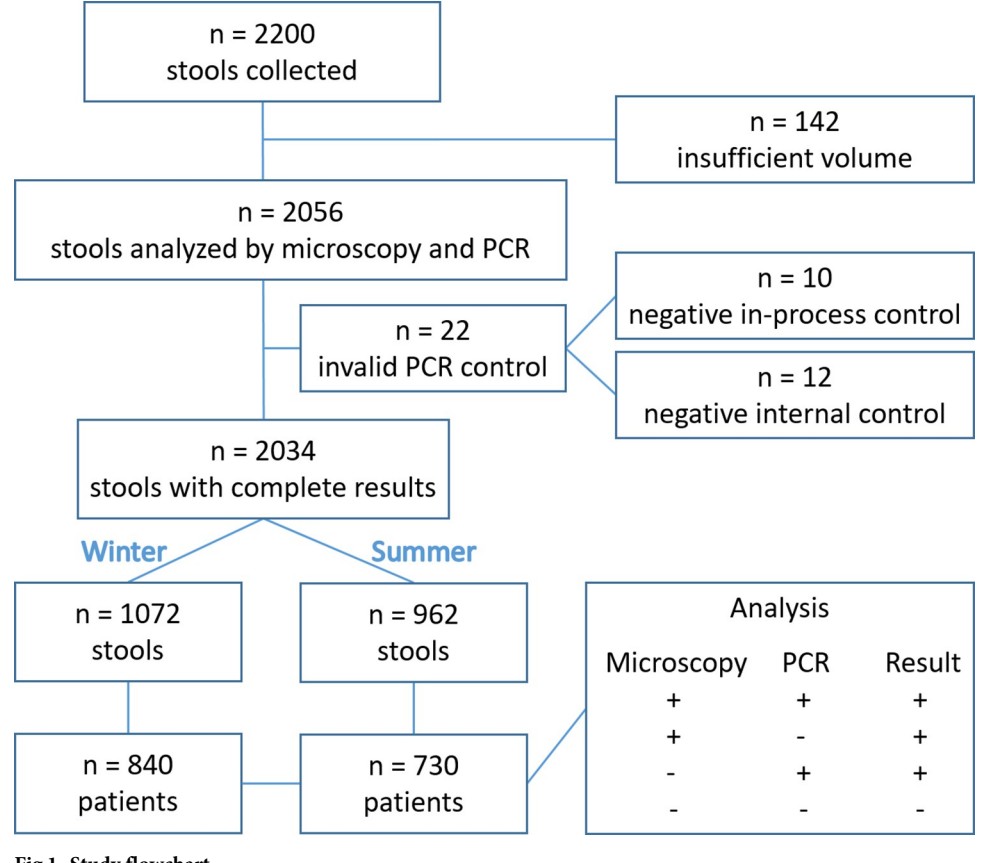

**Fig 1. Study flowchart.**

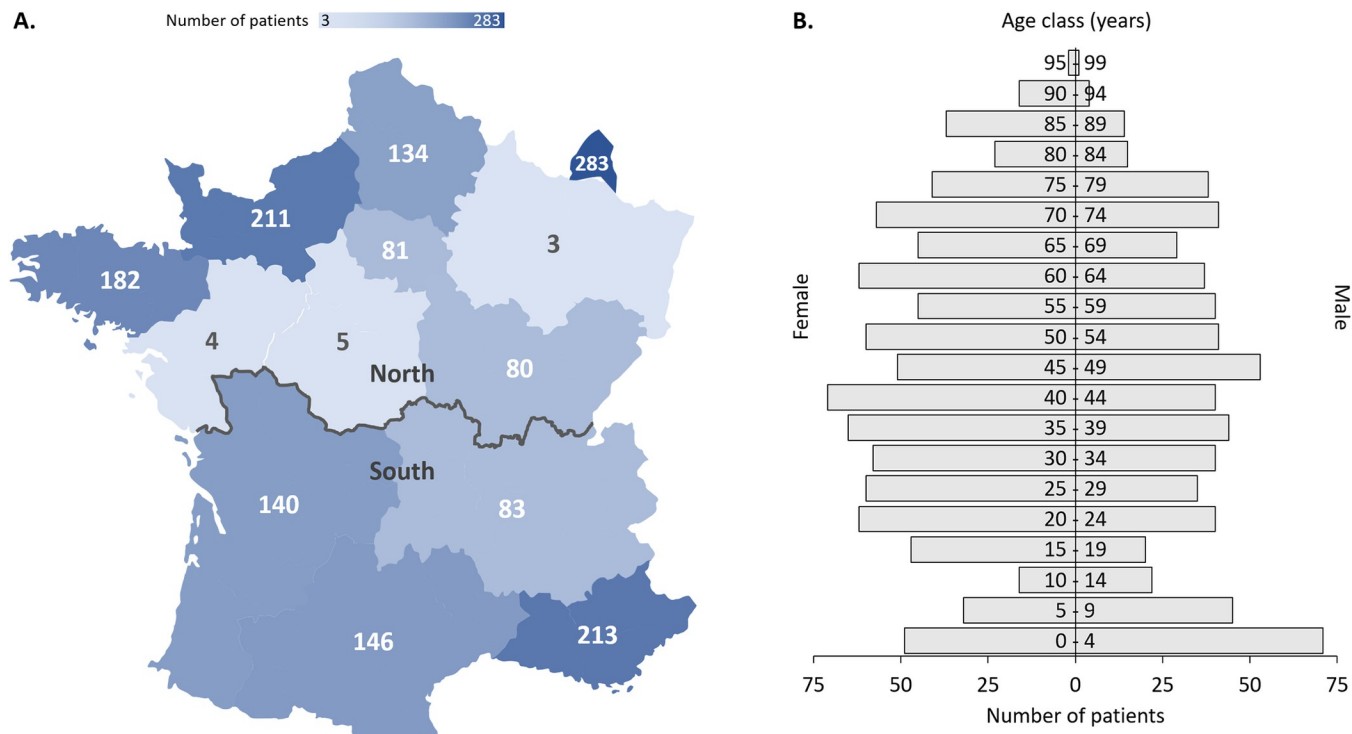

**Fig 2. Study population.** (A) The patient's place of residence at time of sampling is represented on a distribution map downloaded from Servier Medical Art (https://smart.servier.com/?s=map) (n = 1565). The grey line represents separation between North and South areas for statistical analysis. (B) The distribution of patients by sex and age is represented on the corresponding population pyramid (n = 1569).

respectively, followed by *Giardia intestinalis* (1.9%, n = 30), *Cryptosporidium* spp. (1.9%, n = 30), *Enterobius vermicularis* (1.9%, n = 29), and amoeba-like *Entamoeba coli* (1.5%, n = 23), *Endolimax nana* (1.2%, n = 19) and *Entamoeba dispar* (1.1%, n = 17). No *Entamoeba*

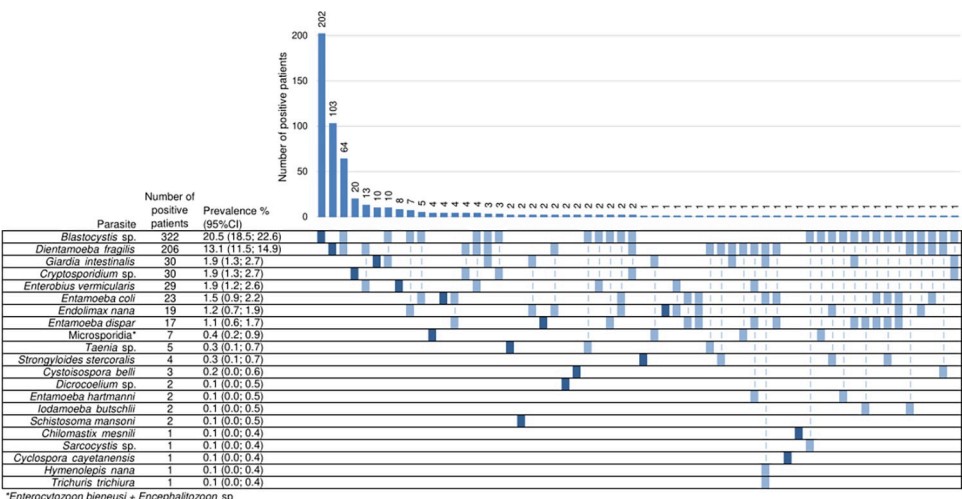

**Fig 3. Prevalence by parasite with coinfections distribution.** Results are presented by patient (n = 1570). Parasites are sorted from most to least prevalent. Dark blue boxes indicate cases with only one parasite detected, the others indicate cases with two or more parasites detected. Coinfections appear vertically with dotted lines to make reading easier. Parasites not shown are equal to zero. CI: confidence interval.

*histolytica* was detected. Some other protozoa were found such as *Cystoisospora belli* (n = 3), *Entamoeba hartmanni* (n = 2), *Iodamoeba butschlii* (n = 2), *Chilomastix mesnili* (n = 1), *Sarcocystis* sp. (n = 1) and *Cyclospora cayetanensis* (n = 1).

Coinfection with two or more parasites occurred in 156 patients (9.9%, 95%CI: 8.5 to 11.5), of whom 126 (8.0%) were coinfected with two parasites, 25 (1.6%) with three parasites, 4 (0.3%) with four parasites, and one (0.1%) with five parasites. This last patient was coinfected with *D. fragilis*, *G. intestinalis*, *E. coli*, *Trichuris trichiura*, and *Hymenolepis nana*. The most frequent associations between two parasites were *Blastocystis* sp./*D. fragilis* (n = 77), followed by *D. fragilis*/*E. vermicularis* (n = 18), *Blastocystis* sp./*G. intestinalis* (n = 15), *Blastocystis* sp./*E. nana* (n = 12) and *Blastocystis* sp./*E. coli* (n = 11) (Fig 3). Significantly more males than females were coinfected (respectively, 13.4% vs 7.3%, p<0.001, S1 Table).

Molecular analysis up to the species or genotype level was performed for the parasites listed below. However, not all were successful for different reasons, including insufficient amount of DNA, degraded DNA or insufficient amplification of the target DNA for sequencing analysis. Of the 322 patients positive for *Blastocystis* sp., 252 isolates were successfully subtyped as follows, from most to least prevalent: ST3 (25.8%), ST4 (25.0%), ST2 (22.2%), ST1 (15.1%), ST7 (7.1%), ST6 (3.6%), and ST8 (1.2%). Likewise, of the 206 patients positive for *D. fragilis*, 143 isolates were successfully genotyped and shown to be mainly of genotype 1 (97.9%). Only three patients (2.1%) were detected with genotype 2. Of the 30 patients positive for *G. intestinalis*, 16 isolates were successfully genotyped into 6 assemblages A, and 10 assemblages B. Among the 30 patients positive for *Cryptosporidium* sp., 14 *Cryptosporidium parvum* and 2 *Cryptosporidium hominis* were identified. The following *C. parvum* subtypes were identified: IIaA15G2R1 (n = 10), IIaA14R1 (n = 1), IIaA17G2R1 (n = 1), IIaA18G3R1 (n = 1), and IIaA22G1 (n = 1). For *C. hominis*, the two genotypes found were IbA9G3 and IbA13G3. In the seven patients harbouring microsporidia, the species involved were *Enterocytozoon bieneusi* (n = 6) and *Encephalitozoon* sp. (n = 1). Unfortunately, *Encephalitozoon* species identification failed but successful genotyping of four *E. bieneusi* isolates showed the presence of genotypes A (n = 1), Peru11 (n = 1), PigEbITS3 (n = 1), and Wildboar3 (n = 1). Of the five positive cases of *Taenia* sp., four were successfully identified as *T. saginata*. The *Sarcocystis* isolate was identified as *Sarcocystis hominis*. *Strongyloides stercoralis* (n = 4), *Dicrocoelium* sp. (n = 2), and *Schistosoma mansoni* (n = 2) were also detected.

## Comparison between PCR and microscopy

A comparison of the number of positive patients obtained by the two types of methods is summarized in Fig 4 and S2 Table. We found that more than half of the cases were diagnosed by PCR alone (blank bars, Fig 4), indicating the higher sensitivity of molecular methods. Some others cases can only be diagnosed by microscopy (dark bars, Fig 4), or were missed by PCR (see *Blastocystis* sp. and *E. vermicularis*). Parasite prevalence according to the diagnostic method used and the number of stool samples per patient is summarized in S3 Table. The overall prevalence was significantly increased with the number of stools collected per patient by microscopy and PCR combined, as well as by microscopy alone or PCR alone (p = 0.001, p<0.001 and p = 0.003, respectively). These results were similar when considering the overall prevalence excluding *Blastocystis* sp. (p = 0.02, p = 0.02 and p = 0.045, respectively), or specifically considering *Blastocystis* sp. (p = 0.002, p<0.001 and p = 0.03, respectively). Parasite carriage was then assessed in the subset of positive patients with 3 stool samples. More than half of the patients were diagnosed with parasites in only one or two of the three samples, both overall and by parasite (Fig 5). Looking at the 21 patients with coinfection in this subset, a second or third sample was often needed to detect all the parasites carried by a patient (S4 Table).

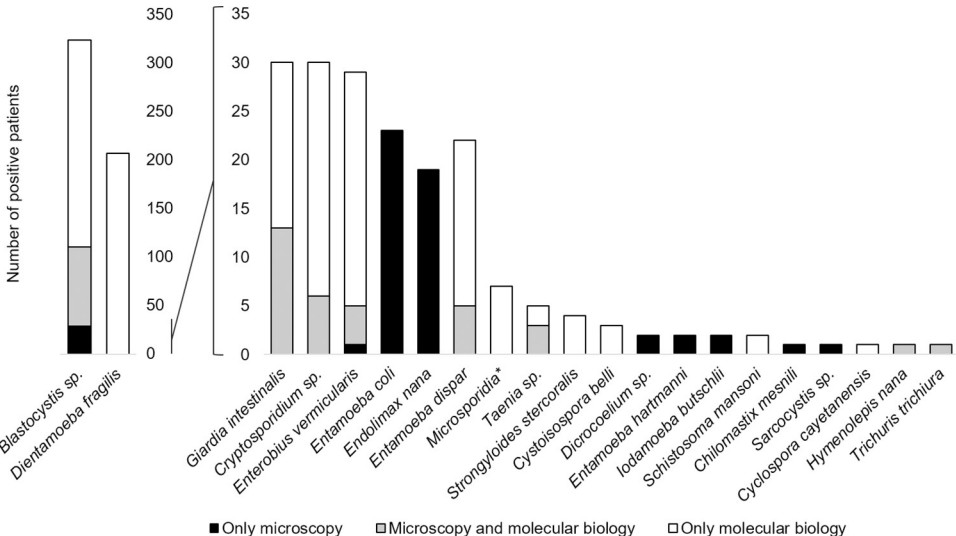

**Fig 4. Comparison of the number of positive patients obtained by microscopy and/or PCR.** Histograms show the number of cases diagnosed by PCR alone (blank bars), by microscopy alone (dark bars), or by both methods (gray bars). *Enterocytozoon bieneusi + Encephalitozoon* sp.

## Factors associated with intestinal parasitic carriage

There was no effect of sex, season or geographical area on overall parasite and microsporidiosis prevalence (Table 1). No significant difference in geographical origin was found between individual parasites from the North and the South of the country except for *G. intestinalis*, which was more prevalent in the South (2.8% *vs* 1.3% in the North, p = 0.04) (Fig 6). *Cryptosporidium* sp. was more prevalent in the summer (3.0% *vs* 1.0% in the winter, p = 0.003), while *Dientamoeba fragilis* was more prevalent in the winter (15.0% *vs* 11.0% in the summer, p = 0.02), as was *Entamoeba coli* (2.0% *vs* 0.8% in the summer; p = 0.048) (Table 1). *Entamoeba coli* was also more prevalent in males than in females (2.2% *vs* 0.9%, respectively, p = 0.03), as was *Entamoeba dispar* (1.9% vs 0.4%, respectively, p = 0.005). Regarding age, *D. fragilis* was significantly more prevalent in children aged 5–14 years old (49.6% *vs* less than 19.2% in other age groups, p<0.001), while *Blastocystis* sp. was less prevalent in children under 5 years old (8.3%

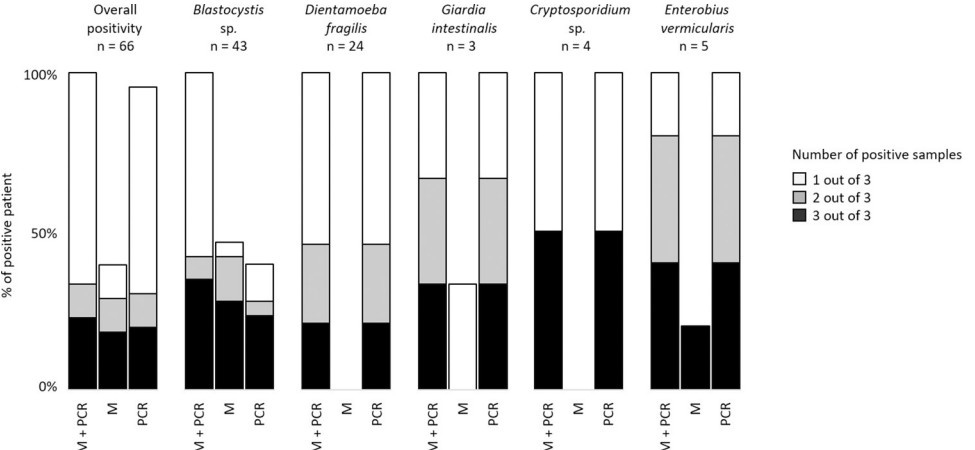

**Fig 5. Number of positive stool samples in the subset of positive patients with 3 stool samples according to the diagnostic method used.** M: Microscopy, PCR: Polymerase Chain Reaction.

**Table 1. Parasite and microsporidiosis prevalence according to sex, geographical origin, season of sampling and age.**

| | Sex | | | Region | | | Season | | | Age (years) | | | | | |
|---|---|---|---|---|---|---|---|---|---|---|---|---|---|---|---|
| | Female (n = 899) | Male (n = 671) | p | North (n = 983) | South (n = 582) | p | Winter (n = 840) | Summer (n = 730) | p | <5 (n = 120) | 5–14 (n = 115) | 15–24 (n = 169) | 25–44 (n = 413) | ≥45 (n = 752) | p |
| Overall positivity | 287 (31.9) | 234 (34.9) | 0.22 | 328 (33.4) | 190 (32.6) | 0.77 | 281 (33.5) | 240 (32.9) | 0.81 | 37 (30.8) | 70 (60.9) | 51 (30.2) | 134 (32.4) | 229 (30.5) | **<0.001** |
| Blastocystis sp. | 171 (19.0) | 151 (22.5) | 0.09 | 211 (21.5) | 110 (18.9) | 0.23 | 171 (20.4) | 151 (20.7) | 0.87 | 10 (8.3) | 26 (22.6) | 28 (16.6) | 87 (21.1) | 171 (22.7) | **0.004** |
| Dientamoeba fragilis | 111 (12.3) | 95 (14.2) | 0.29 | 122 (12.4) | 83 (14.3) | 0.29 | 126 (15.0) | 80 (11.0) | **0.02** | 23 (19.2) | 57 (49.6) | 18 (10.7) | 45 (10.9) | 63 (8.4) | **<0.001** |
| Giardia intestinalis | 15 (1.7) | 15 (2.2) | 0.46 | 13 (1.3) | 16 (2.8) | **0.04** | 11 (1.3) | 19 (2.6) | 0.06 | 1 (0.8) | 5 (4.4) | 3 (1.8) | 6 (1.5) | 15 (2.0) | 0.36 |
| Cryptosporidium sp. | 14 (1.6) | 16 (2.4) | 0.24 | 18 (1.8) | 12 (2.1) | 0.75 | 8 (1.0) | 22 (3.0) | **0.003** | 3 (2.5) | 4 (3.5) | 4 (2.4) | 5 (1.2) | 14 (1.9) | 0.44 |
| Enterobius vermicularis | 16 (1.8) | 13 (1.9) | 0.82 | 18 (1.8) | 11 (1.9) | 0.93 | 19 (2.3) | 10 (1.4) | 0.19 | 4 (3.3) | 8 (7.0) | 6 (3.6) | 5 (1.2) | 6 (0.8) | **<0.001** |
| Entamoeba coli | 8 (0.9) | 15 (2.2) | **0.03** | 18 (1.8) | 5 (0.9) | 0.12 | 17 (2.0) | 6 (0.8) | 0.14 | 0 (0.0) | 3 (2.6) | 1 (0.6) | 8 (1.9) | 11 (1.5) | 0.37 |
| Endolimax nana | 9 (1.0) | 10 (1.5) | 0.38 | 9 (0.9) | 10 (1.7) | 0.16 | 7 (0.8) | 12 (1.6) | 0.96 | 0 (0.0) | 1 (0.9) | 2 (1.2) | 9 (2.2) | 7 (0.9) | 0.30 |
| Entamoeba dispar | 4 (0.4) | 13 (1.9) | **0.005** | 10 (1.0) | 7 (1.2) | 0.73 | 9 (1.1) | 8 (1.1) | **0.048** | 0 (0.0) | 0 (0.0) | 3 (1.8) | 8 (1.9) | 6 (0.8) | 0.18 |
| Microsporidia* | 4 (0.4) | 3 (0.4) | 1.00 | 6 (0.6) | 0 (0.0) | 0.09 | 3 (0.4) | 4 (0.5) | 0.71 | 1 (0.8) | 0 (0.0) | 1 (0.6) | 1 (0.2) | 4 (0.5) | 0.72 |
| Taenia sp. | 2 (0.2) | 3 (0.4) | n.d. | 4 (0.4) | 1 (0.2) | n.d. | 4 (0.5) | 1 (0.1) | n.d. | 0 (0.0) | 0 (0.0) | 1 (0.6) | 2 (0.5) | 2 (0.3) | n.d. |
| Strongyloides stercoralis | 1 (0.1) | 3 (0.4) | n.d. | 2 (0.2) | 2 (0.3) | n.d. | 3 (0.4) | 1 (0.1) | n.d. | 0 (0.0) | 0 (0.0) | 0 (0.0) | 2 (0.5) | 2 (0.3) | n.d. |
| Cystoisospora belli | 2 (0.2) | 1 (0.1) | n.d. | 2 (0.2) | 1 (0.2) | n.d. | 1 (0.1) | 2 (0.3) | n.d. | 0 (0.0) | 0 (0.0) | 1 (0.6) | 0 (0.0) | 2 (0.3) | n.d. |
| Dicrocoelium sp. | 2 (0.2) | 0 (0.0) | n.d. | 2 (0.2) | 0 (0.0) | n.d. | 1 (0.1) | 1 (0.1) | n.d. | 0 (0.0) | 0 (0.0) | 0 (0.0) | 1 (0.2) | 1 (0.1) | n.d. |
| Entamoeba hartmanni | 0 (0.0) | 2 (0.3) | n.d. | 0 (0.0) | 2 (0.3) | n.d. | 0 (0.0) | 2 (0.3) | n.d. | 0 (0.0) | 0 (0.0) | 0 (0.0) | 0 (0.0) | 2 (0.3) | n.d. |
| Iodamoeba butschlii | 1 (0.1) | 1 (0.1) | n.d. | 1 (0.1) | 1 (0.2) | n.d. | 0 (0.0) | 2 (0.3) | n.d. | 0 (0.0) | 0 (0.0) | 1 (0.6) | 1 (0.2) | 0 (0.0) | n.d. |
| Schistosoma sp. | 2 (0.2) | 0 (0.0) | n.d. | 1 (0.1) | 1 (0.2) | n.d. | 1 (0.1) | 1 (0.1) | n.d. | 0 (0.0) | 0 (0.0) | 2 (1.2) | 0 (0.0) | 0 (0.0) | n.d. |
| Chilomastix mesnili | 0 (0.0) | 1 (0.1) | n.d. | 1 (0.1) | 0 (0.0) | n.d. | 1 (0.1) | 0 (0.0) | n.d. | 1 (0.8) | 0 (0.0) | 0 (0.0) | 0 (0.0) | 0 (0.0) | n.d. |
| Sarcocystis sp. | 1 (0.1) | 0 (0.0) | n.d. | 1 (0.1) | 0 (0.0) | n.d. | 0 (0.0) | 1 (0.1) | n.d. | 0 (0.0) | 0 (0.0) | 0 (0.0) | 1 (0.2) | 0 (0.0) | n.d. |
| Cyclospora cayetanensis | 1 (0.1) | 0 (0.0) | n.d. | 1 (0.1) | 0 (0.0) | n.d. | 1 (0.1) | 0 (0.0) | n.d. | 0 (0.0) | 0 (0.0) | 1 (0.6) | 0 (0.0) | 0 (0.0) | n.d. |
| Hymenolepis nana | 0 (0.0) | 1 (0.1) | n.d. | 1 (0.1) | 0 (0.0) | n.d. | 1 (0.1) | 0 (0.0) | n.d. | 0 (0.0) | 1 (0.9) | 0 (0.0) | 0 (0.0) | 0 (0.0) | n.d. |
| Trichuris trichiura | 0 (0.0) | 1 (0.1) | n.d. | 1 (0.1) | 0 (0.0) | n.d. | 1 (0.1) | 0 (0.0) | n.d. | 0 (0.0) | 1 (0.9) | 0 (0.0) | 0 (0.0) | 0 (0.0) | n.d. |

Data are presented as number of patients (percentage). Total number of patients is 1570 for sex and season, 1569 for age group and season, 1569 for age group (one birthdate lacking), and 1565 for geographical origin (five ZIP code lacking, see Fig 2 for the definition of North and South areas). p-values were calculated only for the most commonly represented parasites. To help reading, p-values <0.05 are indicated in bold. n.d.: not determined. *Enterocytozoon bieneusi + Encephalitozoon sp.

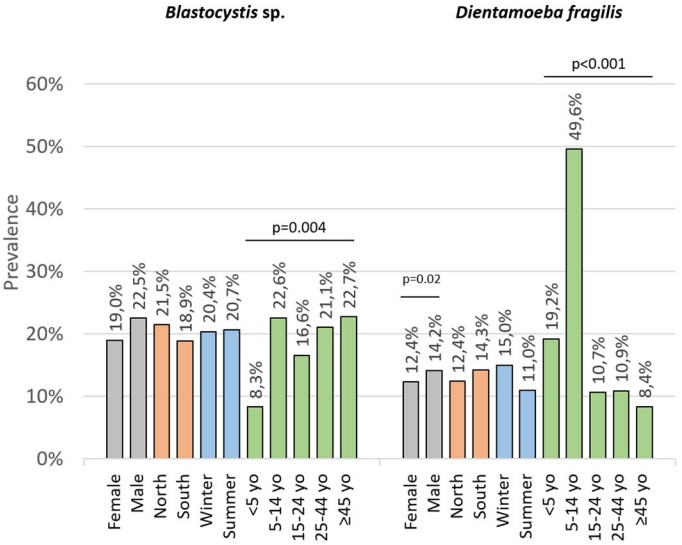

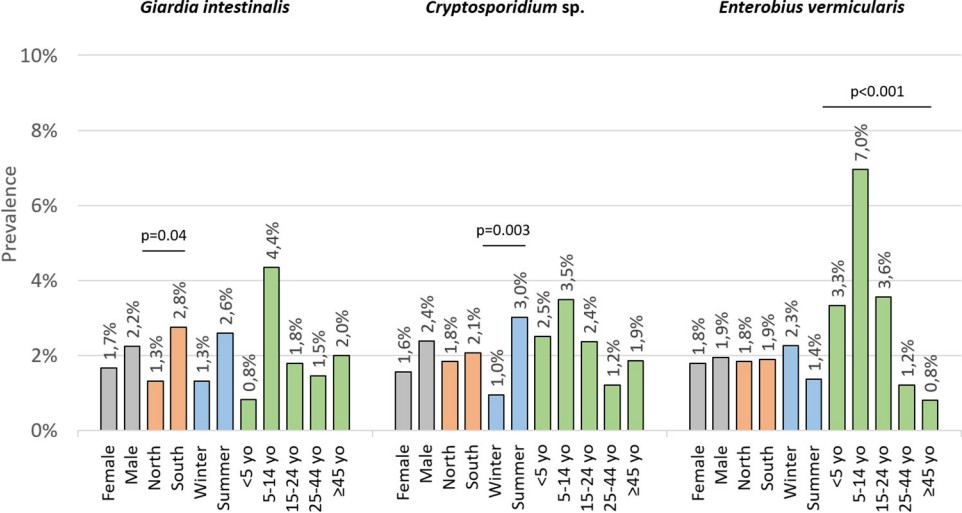

**Fig 6. Prevalence of the five most prevalent parasites according to sex, geographical origin, season of sampling and age.** Results are presented as percentage of patients (n = 1570 for sex and season, n = 1569 for age class, n = 1565 for geographical origin). yo: years old.

*vs* more than 16.6% in other age groups, p = 0.004). *Enterobius vermicularis* was also predominant in the 5- to 14-year-old age class with a prevalence of 7.0%. Multivariable analysis confirmed the differences observed for *Blastocystis* sp. and *D. fragilis* (S5 Table). The parasite prevalence according to the diagnostic method used is summarized in S6 Table. The use of PCR assays increases the proportion of positive patients and strengthens the statistical significance for some variables (*e.g.*, *Giardia* and region or *Cryptosporidium* and season).

## Discussion

Data about the prevalence of intestinal, protozoa, helminths and microsporidiosis in human populations from France and Luxembourg are scarce [21]. The aim of this study was to

provide an epidemiological update of intestinal parasitosis and microsporidiosis in individuals who had been prescribed a stool analysis for the presence of IPs in these two countries. One limitation of this study was the lack of clinical and epidemiological data, which were restricted to the age, sex and residence of the patients. However, this enabled us to harvest at a large geographic scale numerous samples from primary health care patients without distinction and for whom clinicians had requested a search for parasites. Stools were analyzed in a single center by both microscopy and multiplex PCR assays to have the most complete overview possible. Indeed, as an example for *Sarcocystis* sp., none of commercial PCR panel available to date target this parasite, a drawback that reinforces the need to preserve microscopy skills or to push the manufacturers to develop new PCRs for non-included pathogens [22].

Altogether, a parasite prevalence of 33.2% was found in the patients of our study. It is important to emphasize that this high prevalence reflects faecal exposure. The most prevalent parasite was *Blastocystis* sp. (20.5%). This is in line with recent studies in France, where the prevalence of *Blastocystis* sp. ranged from 10.5% to 18.1% [2,5,11]. We observed no sex or geographical differences, but children under five years old were less frequently positive for *Blastocystis* sp. compared to other age groups (8.3% *vs* more than 16.6%). The most frequently found subtypes of *Blastocystis* sp. in our study were, in order, ST3, ST4, ST2 and ST1, all of which being frequently reported in humans, including France and other European countries [2,5,23,24]. As previously reported, both ST6 and ST7, considered to be avian STs, were rarer [2,5]. Interestingly, three patients were diagnosed with ST8, which is frequently reported in monkeys but rarely in humans, even though it was reported in Italy and recently observed in abundance in wastewater samples in Denmark [23,24]. *Dientamoeba fragilis* was the second most commonly detected parasite (13.1%). This result corroborate the hypothesis of a South to North gradient of prevalence, as *D. fragilis* was reported to varies from about 1% in Spain and Italy to more than 60% in Northern Europe [25]. The significantly higher prevalence in children under the age of 14 (49.6%) observed in this study is in line with a recent study performed in France [26]. We also found a seasonal pattern of *D. fragilis* carriage with a higher prevalence in winter (15.0%) than in summer (11.0%). Among the *D. fragilis* genotypes, only three patients from different geographical areas were detected with genotype 2, which is rarely reported [25]. Even though the presence of *Blastocystis* sp. and *D. fragilis* was reported in individuals with intestinal inflammation or irritable bowel syndrome, recent studies associated them to healthier microbiomes, suggesting a beneficial effect [25,27–31].

Prevalences were lower for pathogenic parasites such as *G. intestinalis*, *Cryptosporidium* sp. and *E. vermicularis* with an equal prevalence of 1.9%. Frequent isolation of *G. intestinalis* is consistent with previous reports and, to a lesser extent, that of *E. vermicularis* [2,3,8,9]. Five cases of pinworms were diagnosed by microscopy, the others by PCR only. As previously reported, there is a prevalence peak in the 5- to 14-year-old age class, which in our study was up to 7.0%. *Dientamoeba fragilis* was detected in 62.1% of patients infected by *E. vermicularis*. This high coinfection rate has already been reported leading to hypothesis about the carriage of *D. fragilis* by *E. vermicularis* ova [25]. For *G. intestinalis*, a significantly higher prevalence was found in the South than in the North, but without a clear explanation. Genotyping identified in our study were the two zoonotic assemblages A and B, mostly found in humans but also identified in various mammals [32]. In most reports, especially in Europe, assemblage B is more frequent in Human than assemblage A [32]. Interestingly, *Cryptosporidium* sp. was identified as frequently as *G. intestinalis* and *E. vermicularis*, which confirms the status of *Cryptosporidium* sp. as a common pathogenic parasite in the general population. Our study also confirmed the seasonality of cryptosporidiosis, which had a 3-fold higher prevalence in summer than in winter (3.0% *vs* 1.0%), probably as the result of an increased consumption of raw vegetables and exposure to recreational waters, as previously reported [6]. In the

*Cryptosporidium* species, *C. parvum* is known to be zoonotic whereas *C. hominis* seems to be more specific, but not exclusively, to humans. Only these two species were detected in our study, which is consistent with the Costa *et al.* study, who showed that 96% of human cases between 2017 and 2019 were caused by *C. parvum* (72%) and *C. hominis* (24%) [6]. The genotyping results of *Cryptosporidium* from our sample populations are also consistent with the known European epidemiology of *Cryptosporidium* [6,33–35]. Like for *Cryptosporidium sp.*, microsporidia should be searched in primary health care patients because it is not restricted to immunocompromised patients as highlighted by Danish and Swedish outbreaks [36–38]. A prevalence of 0.4% was found in our study making it a more common pathogen than *Taenia* sp. or *S. stercoralis*, for example. The genotypes we identified belong to the zoonotic Group 1 of *E. bieneusi*, which is consistent with the environmental reservoir and the faecal-oral route of transmission [39].

It is noteworthy that a relatively high number of patients had parasite coinfection. Almost 10% of patients carried two or more parasites, with certain preferential associations such as *Blastocystis* sp./*D. fragilis*, *D. fragilis*/*E. vermicularis* and *Blastocystis* sp./*G. intestinalis*, or harboured non-pathogenic intestinal amoebae *(E. coli, E. nana, E. dispar, Iodamoeba butschlii, Entamoeba hartmanni)* indicating a direct or indirect faecal exposure. The age group 5–14 years was significantly more coinfected than others, with mainly the *Blastocystis* sp./*D. fragilis* association, both parasites showing the same difference in age class repartition. These findings show that multiple infections are not rare and could be even more common when enteropathogenic bacteria and viruses are taken into consideration, as described in recent studies [40].

Finally, a comparison between PCR and microscopy, although our study was not designed for it, shows some interesting preliminary data. First, as expected, the use of multiplex PCR-based diagnostics increases the proportion of positive patients. Second, the number of positive patients from our study increases significantly with the multiplication of samples analysed per patients, by using microscopy methods as well as by PCR. Thus, our results suggest that the increased sensitivity of molecular assays does not exempt to repeat sampling for parasitological testing. Further dedicated studies are required to address that question and purpose recommendations in the number of samples to test when using PCR alone.

In conclusion, a high prevalence of intestinal parasites was found in France and Luxembourg with almost one third of patients being positive for at least one parasite. Even though the most frequently detected parasites were *Blastocystis* sp. and *D. fragilis*, the high prevalence of pathogenic intestinal parasites (about 7%) highlights the importance of parasitic screening, even in high economic level countries. In addition, the high frequency of coinfection with two or more parasites should encourage the use of the most exhaustive testing methodology, probably by using a combination between microscopy and PCR. The detection of parasites, of proven pathogenicity or not, remains a marker of faecal exposure and results should be interpreted accordingly. Finally, clinicians should be familiar with this epidemiology and be aware that industrialized countries are also affected by intestinal parasitic infection.

## Supporting information

**S1 Table. Parasites coinfection by microscopy and/or molecular biology according to sex, geographical origin, season of sampling and age.**
(DOCX)

**S2 Table. Prevalence by parasite according to the diagnostic method used.**
(DOCX)

**S3 Table. Parasite and microsporidiosis prevalence by microscopy and molecular biology according to the number of stool samples per patient.**
(DOCX)

**S4 Table. Number of stool samples required to detect all coinfections in the subset of patients with three stool samples according to the diagnostic method used.**
(DOCX)

**S5 Table. Multivariable analysis of Blastocystis sp. and Dientamoeba fragilis carriage.**
(DOCX)

**S6 Table. Parasites and microsporidia prevalence by microscopy and molecular biology according to sex, geographical origin, season of sampling and age.**
(DOCX)

**S1 Data. Deidentified raw data used in the analysis.**
(XLSX)

## Acknowledgments

We thank all skilled microscopic operators that prepared and read all samples especially Gaelle Castan, Gwenaelle Guillet, Nathalie Joaquim, Xavier Sansico and Régine Serieys; and all students that helped to prepare samples (Joao Almeida, Chloé Belot, Issam Ben Rhouma, Farah Boughanmi, Apolline Riffard and Antoine Tranquy).

## Author Contributions

**Conceptualization:** Maxime Moniot, Céline Nourrisson, Philippe Poirier.

**Data curation:** Maxime Moniot, Céline Lambert.

**Investigation:** Maxime Moniot, Eloïse Bailly, Patricia Combes.

**Methodology:** Maxime Moniot, Céline Nourrisson, Philippe Poirier.

**Supervision:** Philippe Poirier.

**Writing – original draft:** Maxime Moniot.

**Writing – review & editing:** Maxime Moniot, Céline Nourrisson, Eloïse Bailly, Céline Lambert, Patricia Combes, Philippe Poirier.

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
