## [Decision Letter · Decision Letter 0]

26 Aug 2024

Dear Mr. Poirier,

Thank you very much for submitting your manuscript "Current status of intestinal parasitosis in industrialized countries: results from a prospective study in France and Luxembourg." for consideration at PLOS Neglected Tropical Diseases. As with all papers reviewed by the journal, your manuscript was reviewed by members of the editorial board and by several independent reviewers. In light of the reviews (below this email), we would like to invite the resubmission of a significantly-revised version that takes into account the reviewers' comments. 

We cannot make any decision about publication until we have seen the revised manuscript and your response to the reviewers' comments. Your revised manuscript is also likely to be sent to reviewers for further evaluation.

Sincerely,

María Victoria Periago

Academic Editor

Susan Madison-Antenucci

Section Editor

Reviewer's Responses to Questions

**Key Review Criteria Required for Acceptance?**

**Methods**

-Are the objectives of the study clearly articulated with a clear testable hypothesis stated?

-Is the study design appropriate to address the stated objectives?

-Is the population clearly described and appropriate for the hypothesis being tested?

-Is the sample size sufficient to ensure adequate power to address the hypothesis being tested?

-Were correct statistical analysis used to support conclusions?

-Are there concerns about ethical or regulatory requirements being met?

Reviewer #1: (No Response)

Reviewer #2: See "Summary and General Comments" box

**Results**

-Does the analysis presented match the analysis plan?

-Are the results clearly and completely presented?

-Are the figures (Tables, Images) of sufficient quality for clarity?

Reviewer #1: (No Response)

Reviewer #2: See "Summary and General Comments" box

**Conclusions**

-Are the conclusions supported by the data presented?

-Are the limitations of analysis clearly described?

-Do the authors discuss how these data can be helpful to advance our understanding of the topic under study?

-Is public health relevance addressed?

Reviewer #1: (No Response)

Reviewer #2: See "Summary and General Comments" box

**Editorial and Data Presentation Modifications?**

Reviewer #1: (No Response)

Reviewer #2: See "Summary and General Comments" box

**Summary and General Comments**

Reviewer #1: Line 112, preserved or unpreserved samples?

Line 158, Sarcocystis was not in the multiplex panel?

Line 197, Microscopy, PCR? Presumably combined please state.

Line 236, Strongyloides stercoralis is a species, what do the authors mean by species identification failed in that case?

Line 260 and further, the authors seem to ignore the accumulating evidence that Blastocystis and Dientamoeba fragilis are indicators for a healthy microbiome rather than causing disease. This should at least be discussed. 

Line 380, rephrase intestinal parasitic infections

Reviewer #2: I would like to thank the editor who offered me this evaluation work. 

Since I am not a native English speaker, my proposed changes will only concern the work presented and not the grammar or syntax. 

In this manuscript, the authors aimed to provide an epidemiological update on intestinal parasites found in community care in France and Luxembourg.

The paper is clear and well written. This study is of interest because there is limited data in the literature about the prevalence of intestinal parasite in these two countries.

Major comment:

One major comment concerns the lack of data on diagnostic methods and the absence of exploitation of Microscopy/ PCR data.

In the material and method section, the authors present no data on the articulation between the two types of diagnostic methods based on either microscopy or PCR. How were mismatches between the 2 types of test handled (patient PCR(+)/ microscopy(-) or vice versa)? Did one technique correct the results of the other?

The flow chart doesn't provide any information on this subject, and seems to put the emphasis on PCR, since samples with an invalid PCR control were excluded from the study, whereas microscopy should have been available.

Even if this is an epidemiological study, the proportion of each diagnostic method (PCR or microscopy) that enabled detection of the various parasites must be clearly shown for each microorganism. In the same way as the authors investigated the influence of the variables gender, region, season or age on positive patients, I believe it is necessary to do the same with the variables "use of microscopic diagnosis" or "use of PCR diagnosis".

Given that each stool was analyzed by PCR and microscopy, a comparison of the number of positives obtained by the two types of method would make it possible to identify the gain in detection linked to PCR, as suggested by the authors in line 80 ("The use of molecular assays shows that intestinal parasites are underestimated in industrialized countries"). This has not been done and is missing from the study.

Finally, a comparison between PCR and microscopy could give an initial idea of the performance of the Biosynex test, for which no publication exists to date.

Minor comments:

1) Since Microsporidia belong to the fungi kingdom, it is theoretically necessary to add microsporidia infections whenever the term "intestinal parasite" or "intestinal parasitic infection" is used. In the title, add "microsporidiosis": “Current status of intestinal parasitosis and microsporidiosis…”

For the rest of the text, it's a bit cumbersome to systematically add the term "microsporidia", but it should be reiterated at the start of each important new section, in the conclusions and in the abstract.

2) Line 78: Define the term "pathogenic" in materials and methods. Authors should identify by name the species they have considered to be pathogenic.

3) Line 91: Add the Calderao et al study from Italy (Calderaro et al. Intestinal parasitoses in a tertiary-care hospital located in a non-endemic setting during 2006-2010. BMC Infect Dis. 2014, doi: 10.1186/1471-2334-14-264. PMID: 24886502)

4) Line 112-113: Stool samples were collected prospectively from all regions of France and Luxembourg. No description of pre-analytical conditions was provided by the authors (storage, preservation solution, time to microscopic analysis). Add these information in the material and methods section.

5) Line 114 -115: Add “adult-stage helminths” and “parasite and microsporidia DNA” to the sentence “were investigated for the presence of ova, cysts, larvae, and vegetative forms.

6) Line 139: According to the manufacturer’s instructions, the AMPLIQUICK® Protozoans assay targets Encephalitozoon spp. (not Encephalitozoon intestinalis).

7) Line 142-143 : According to the manufacturer’s instructions, the AMPLIQUICK® Helminths assay targets Diphyllobothrium latum and Schistosoma mansoni (not Diphyllobothrium sp. and Schistosoma sp.)

8) Line 206: Results are presented per patient. The same patient may therefore have several stools. The authors do not describe how they considered a patient with different parasites isolated from different stool samples (coinfections? independent parasites?). Add these information in the material and methods section. 

9) Line 211: It's a pity that the authors didn't try to associate parasitic co-occurrences with the data collected (gender, region, season or age, method of diagnosis).

10) Line 240-247: It is also a pity that a multivariate analysis was not carried out to identify the parameters that best explain the "Parasite detection" variable. For example, for Giardia, it would be interesting to analyze the explanatory variables "Region (p=0.04)", "Season (p=0.06)" or "Age 5-14" to identify which of these parameters best explains positivity for this parasite. The same can be done for any other parasites detected.

11) Line 242: The 2 prevalence values have been reversed. Replace « 1.3% vs 2.8% in the North » with « 2.8% vs 1.3% in the North».

12) Line 263: The sentence "The aim of this study was to make an epidemiological update of intestinal parasitosis in Europe..." needs to be reworded. The term "Europe" is geographically too broad for an epidemiological study carried out in only 2 neighbouring countries (one of which has a small surface area).

13) Line 272: To the sentence “a drawback that reinforces the need to conserve microscopic skills” could be added “or to push the manufacturers to develop new PCR for these targets”

14) Line 295: Add the Tchamwa Bamini et al. study from France (Prevalence of D. fragilis: 6.5%) 

(Tchamwa Bamini et al. No evidence of pathogenicity of Dientamoeba fragilis following detection in stools: A case-control study. Parasite. 2024;31:40. doi: 10.1051/parasite/2024041. Epub 2024 Jul 24. PMID: 39052010)

15) Line 297: The significantly higher prevalence in children under the age of 14 observed in this study is in line with the results of the Tchamwa Bamini et al. study.

16) Line 344: It would be interesting to know how Enterobius vermicularis cases were diagnosed (PCR or microscopy).

17) Line 372 (and line 66 and 81): I think the sentence “The high prevalence of intestinal parasites underlines the importance of investigating gastrointestinal disorders through parasite examination, even in developped countries” needs to be amended. Indeed, almost four-fifths of identified parasites have no proven pathogenicity, and this sentence makes a clumsy shortcut between this very high overall prevalence and the need for diagnosis linked to the presence of gastrointestinal disorders. For greater clarity, I suggest: "The high prevalence of pathogenic intestinal parasites (about 7%) underlines the importance of investigating gastrointestinal disorders through parasite examination, even in developped countries".

PLOS authors have the option to publish the peer review history of their article (what does this mean?). If published, this will include your full peer review and any attached files.

Reviewer #1: No

Reviewer #2: No
---

## [Editor Report · Decision Letter 1]

4 Nov 2024

PNTD-D-24-01049R1Current status of intestinal parasitosis and microsporidiosis in industrialized countries: results from a prospective study in France and Luxembourg.PLOS Neglected Tropical Diseases Dear Dr. Poirier, Thank you for submitting your manuscript to PLOS Neglected Tropical Diseases. After careful consideration, we feel that it has merit but does not fully meet PLOS Neglected Tropical Diseases's publication criteria as it currently stands. Therefore, we invite you to submit a revised version of the manuscript that addresses the points raised during the review process. Please submit your revised manuscript within 30 days Dec 04 2024 11:59PM. If you will need more time than this to complete your revisions, please reply to this message or contact the journal office at plosntds@plos.org. Please include the following items when submitting your revised manuscript:*
A rebuttal letter that responds to each point raised by the editor and reviewer(s). You should upload this letter as a separate file labeled 'Response to Reviewers'. This file does not need to include responses to any formatting updates and technical items listed in the 'Journal Requirements' section below.*
A marked-up copy of your manuscript that highlights changes made to the original version. You should upload this as a separate file labeled 'Revised Manuscript with Track Changes'.*
An unmarked version of your revised paper without tracked changes. You should upload this as a separate file labeled 'Manuscript'. If you would like to make changes to your financial disclosure, competing interests statement, or data availability statement, please make these updates within the submission form at the time of resubmission. Guidelines for resubmitting your figure files are available below the reviewer comments at the end of this letter. We look forward to receiving your revised manuscript. Kind regards, María Victoria PeriagoAcademic EditorPLOS Neglected Tropical Diseases Susan Madison-AntenucciSection EditorPLOS Neglected Tropical Diseases

Shaden Kamhawi

co-Editor-in-Chief

Paul Brindley

co-Editor-in-Chief

 **Journal Requirements:** **Additional Editor Comments (if provided):** The authors have modified and revised the manuscript according to the reviewers suggestions. Nonetheless, I have revised the re-submitted manuscript and have some minor comments that need to be addressed before we can accept the study for publication.

The population of the study is unclear, throughout the manuscript the authors refer to the study population as primary care patients, primary care, community based care patients, outpatients, community care, etc. This needs to be uniform throughout and clearly explained in the method section, it's still not clear to me where the stool samples came from.

Also, in some sections the authors mention co-occurrence of parasites and in other co-infection, I suggest using co-infection.

Finally, the discussion is too long and drawn out, please try to shorten in by providing more concise comparison with other studies.

I have also added specific comments and corrections to the attached file.**Reviewers' comments:**   **Figure resubmission:** While revising your submission, please upload your figure files to the Preflight Analysis and Conversion Engine (PACE) digital diagnostic tool, https://pacev2.apexcovantage.com/. PACE helps ensure that figures meet PLOS requirements. To use PACE, you must first register as a user. Registration is free. Then, login and navigate to the UPLOAD tab, where you will find detailed instructions on how to use the tool. If you encounter any issues or have any questions when using PACE, please email PLOS at figures@plos.org. Please note that Supporting Information files do not need this step. If there are other versions of figure files still present in your submission file inventory at resubmission, please replace them with the PACE-processed versions. **Reproducibility:** To enhance the reproducibility of your results, we recommend that authors of applicable studies deposit laboratory protocols in protocols.io, where a protocol can be assigned its own identifier (DOI) such that it can be cited independently in the future. Additionally, PLOS ONE offers an option to publish peer-reviewed clinical study protocols. Read more information on sharing protocols at https://plos.org/protocols?utm_medium=editorial-email&utm_source=authorletters&utm_campaign=protocols

---

## [Editor Report · Decision Letter 2]

2 Dec 2024

Dear Mr. Poirier,

We are pleased to inform you that your manuscript 'Current status of intestinal parasitosis and microsporidiosis in industrialized countries: results from a prospective study in France and Luxembourg.' has been provisionally accepted for publication in PLOS Neglected Tropical Diseases.

Best regards,

María Victoria Periago

Academic Editor

Susan Madison-Antenucci

Section Editor

Shaden Kamhawi

co-Editor-in-Chief

Paul Brindley

co-Editor-in-Chief

---

## [Editor Report · Acceptance letter]

14 Dec 2024

Dear Mr. Poirier,

We are delighted to inform you that your manuscript, "Current status of intestinal parasitosis and microsporidiosis in industrialized countries: results from a prospective study in France and Luxembourg.," has been formally accepted for publication in PLOS Neglected Tropical Diseases.

Best regards,

Shaden Kamhawi

co-Editor-in-Chief

Paul Brindley

co-Editor-in-Chief
